# Ultra-low damping insulating magnetic thin films get perpendicular

Lucile Soumah[1], Nathan Beaulieu[2], Lilia Qassym[3], Cécile Carrétéro[1], Eric Jacquet[1], Richard Lebourgeois[3], Jamal Ben Youssef[2], Paolo Bortolotti[1], Vincent Cros[1] & Abdelmadjid Anane [1]

A magnetic material combining low losses and large perpendicular magnetic anisotropy (PMA) is still a missing brick in the magnonic and spintronic fields. We report here on the growth of ultrathin Bismuth doped $Y_3Fe_5O_{12}$ (BiYIG) films on $Gd_3Ga_5O_{12}$ (GGG) and substituted GGG (sGGG) (111) oriented substrates. A fine tuning of the PMA is obtained using both epitaxial strain and growth-induced anisotropies. Both spontaneously in-plane and out-of-plane magnetized thin films can be elaborated. Ferromagnetic Resonance (FMR) measurements demonstrate the high-dynamic quality of these BiYIG ultrathin films; PMA films with Gilbert damping values as low as $3 \times 10^{-4}$ and FMR linewidth of 0.3 mT at 8 GHz are achieved even for films that do not exceed 30 nm in thickness. Moreover, we measure inverse spin hall effect (ISHE) on Pt/BiYIG stacks showing that the magnetic insulator's surface is transparent to spin current, making it appealing for spintronic applications.

[1] Unité Mixte de Physique CNRS, Thales, Univ. Paris-Sud, Université Paris Saclay, 91767 Palaiseau, France. [2] LABSTICC, UMR 6285 CNRS, Université de Bretagne Occidentale, 29238 Brest, France. [3] Thales Research and Technology, Thales, 91767 Palaiseau, France. Correspondence and requests for materials should be addressed to A.A. (email: madjid.anane@u-psud.fr)

pintronics exploits the electron's spin in ferromagnetic transition metals for data storage and data processing. Interestingly, as spintronics codes information in the angular momentum degrees of freedom, charge transport and therefore the use of conducting materials is not a requirement, opening thus electronics to insulators. In magnetic insulators (MI), pure spin currents are described using excitation states of the ferromagnetic background named magnons (or spin waves). Excitation, propagation and detection of magnons are at the confluent of the emerging concepts of magnonics[1,2], calori-tronics[3], and spin-orbitronics[4]. Magnons, and their classical counterpart, the spin waves (SWs), can carry information over distances as large as millimeters in high-quality thick YIG films, with frequencies extending from the GHz to the THz regime[5–7]. The main figure of merit for magnonic materials is the Gilbert damping $\alpha$[1,5,8] which has to be as small as possible. This makes the number of relevant materials for SW propagation quite limited and none of them has yet been found to possess a large enough perpendicular magnetic anisotropy (PMA) to induce spontaneous out-of-plane magnetization. We report here on the Pulsed Laser Deposition (PLD) growth of ultra-low loss MI nanometers-thick films with large PMA: Bi substituted Yttrium Iron Garnet ($Bi_xY_{3-x}Fe_5O_{12}$ or BiYIG) where tunability of the PMA is achieved through epitaxial strain and Bi doping level. The peak-to-peak FMR linewidth (that characterize the losses) can be as low as $\mu_0\Delta H_{pp} = 0.3$ mT at 8 GHz for 30 nm thick films. This material thus opens new perspectives for both spintronics and magnonics fields as the SW dispersion relation can now be easily tuned through magnetic anisotropy without the need of a large bias magnetic field. Moreover, energy efficient data storage devices based on magnetic textures existing in PMA materials like magnetic bubbles, chiral domain walls, and magnetic skyrmions would benefit from such a low loss material for efficient operation[9].

The study of micron-thick YIG films grown by liquid phase epitaxy (LPE) was among the hottest topics in magnetism few decades ago. At this time, it has been already noticed that unlike rare earths (Thulium, Terbium, Dysprosium …) substitutions, Bi substitution does not overwhelmingly increase the magnetic losses[10,11] even though it induces high uniaxial magnetic aniso-tropy[12–14]. Very recently, ultra-thin MI films showing PMA have been the subject of an increasing interest:[15,16] $Tm_3Fe_5O_{12}$ or $BaFe_{12}O_{19}$ (respectively a garnet and an hexaferrite) have been used to demonstrate spin-orbit-torque magnetization reversal using a Pt over-layer as a source of spin current[4,17,18]. However, their large magnetic losses prohibit their use as a spin-wave medium (reported value of $\mu_0\Delta H_{pp}$ of TIG is 16.7 mT at 9.5 GHz)[19]. Hence, whether it is possible to fabricate ultra-low loss thin films with a large PMA that can be used for both magnonics and spintronics applications remains to be demonstrated. Indeed, not only low losses are important for long range spin wave propagation but they are also necessary for spin transfer torque oscillators (STNOs) as the threshold current scales with the Gilbert damping[20].

In the quest for the optimal material platform, we explore here the growth of Bi doped YIG ultra-thin films using PLD with different substitution; $Bi_xY_{3-x}IG$ ($x = 0.7, 1,$ and $1.5$) and having a thickness ranging between 8 and 50 nm. We demonstrate fine tuning of the magnetic anisotropy using epitaxial strain and measure ultra low Gilbert damping values ($\alpha = 3 \times 10^{-4}$) on ultrathin films with PMA.

## Results
### Structural and magnetic characterizations.
The two substrates that are used are gallium gadolinium garnet (GGG), which is best

lattice matched to pristine YIG and substituted GGG (sGGG) which is traditionally used to accommodate substituted YIG films for photonics applications. The difference between Bi and Y ionic radii ($r_{Bi} = 113$ pm and $r_Y = 102$ pm)[21] leads to a linear increase of the $Bi_xY_{3-x}IG$ bulk lattice parameter with Bi content (Fig. 1a, b). In Fig. 1, we present the ($2\theta$–$\omega$) X-ray diffraction patterns (Fig. 1c, d) and reciprocal space maps (RSM) (Fig. 1e, f) of BiYIG on sGGG (111) and GGG(111) substrates, respectively. The presence of (222) family peaks in the diffraction spectra shown in Fig. 1b, c is a signature of the films' epitaxial quality and the presence of Laue fringes attests the coherent crystal structure existing over the whole thickness. As expected, all films on GGG are under compressive strain, whereas films grown on sGGG exhibit a transition from a tensile (for $x = 0.7$ and 1) towards a compressive ($x = 1.5$) strain. Reciprocal space mapping of these BiYIG samples shown in Fig. 1e, f evidences the pseudomorphic nature of the growth for all films, which confirms the good epitaxy.

The static magnetic properties of the films have been characterized using SQUID magnetometry, Faraday rotation measurements and Kerr microscopy. As the Bi doping has the effect of enhancing the magneto-optical response[22–24], we measure on average a large Faraday rotation coefficients reaching up to $\theta_F = -3°\mu m^{-1}$ @ 632 nm for $x = 1$ Bi doping level and 15 nm film thickness. Chern et al.[25] performed PLD growth of $Bi_xY_{3-x}IG$ on GGG and reported an increase of $\theta_F = -1.9°\mu m^{-1}$ per Bi substitution $x$@ 632 nm. The Faraday rotation coefficients we find are slightly larger and may be due to the much lower thickness of our films as $\theta_F$ is also dependent on the film thickness[26]. The saturation magnetization ($M_s$) remains constant for all Bi content (see Table 1) within the 10% experimental errors. We observe a clear correlation between the strain and the shape of the in-plane and out-of-plane hysteresis loops reflecting changes in the magnetic anisotropy. While films under compressive strain exhibit in-plane anisotropy, those under tensile strain show a large out-of-plane anisotropy that can eventually lead to an out-of-plane easy axis for $x = 0.7$ and $x = 1$ grown on sGGG. The transition can be either induced by changing the substrate (Fig. 2a) or the Bi content (Fig. 2b) since both act on the misfit strain. We ascribe the anisotropy change in our films to a combination of magneto-elastic anisotropy and growth-induced anisotropy, this later term being the dominant one (see Supplementary Note 1).

In Fig. 2c, we show the magnetic domains structures at remanance observed using polar Kerr microscopy for $Bi_1Y_2IG$ films after demagnetization: μm-wide maze-like magnetic domains demonstrates unambiguously that the magnetic easy axis is perpendicular to the film surface. We observe a decrease of the domain width ($D_{width}$) when the film thickness ($t_{film}$) increases as expected from magnetostatic energy considerations. In fact, as $D_{width}$ is several orders of magnitude larger than $t_{film}$, a domain wall energy of $\sigma_{DW} \sim 0.7$ and $0.65$ mJ m$^{-2}$ (for $x = 0.7$ and 1 Bi doping) can inferred using the Kaplan and Gerhing model[27] (the fitting procedure is detailed in the Supplementary Note 2).

### Dynamical characterization and spin transparency.
The most striking feature of these large PMA films is their extremely low magnetic losses that we characterize using Ferromagnetic Resonance (FMR) measurements. First of all, we quantify by in-plane FMR the anisotropy field $H_{KU}$ deduced from the effective magnetization ($M_{eff}$): $H_{KU} = M_S - M_{eff}$ (the procedure to derive $M_{eff}$ from in-plane FMR is presented in Supplementary Note 3). $H_{KU}$ values for BiYIG films with different doping levels grown on various substrates are summarized in Table 1. As expected from out-of-plane hysteresis curves, we observe different signs for $H_{KU}$. For spontaneously out-of-plane magnetized samples, $H_{KU}$ is

positive and large enough to fully compensate the demagnetizing field while it is negative for in-plane magnetized films. From these results, one can expect that fine tuning of the Bi content allows fine tuning of the effective magnetization and consequently of the FMR resonance conditions. We measure magnetic losses on a 30 nm thick $Bi_1Y_2IG//sGGG$ film under tensile strain with PMA (Fig. 3a). We use the FMR absorption line shape to extract the peak-to-peak linewidth ($\Delta H_{pp}$) at different out-of-plane angle for a 30 nm thick perpendicularly magnetized $Bi_1Y_2IG//sGGG$ film at

8 GHz (Fig. 3b). This yields an optimal value of $\mu_0\Delta H_{pp}$ as low as 0.3 mT (Fig. 3c) for 27° out-of-plane polar angle. We stress here that state-of-the-art PLD grown YIG//GGG films exhibit similar values for $\Delta H_{pp}$ at such resonant conditions[28]. This angular dependence of $\Delta H_{pp}$ that shows pronounced variations at specific angle is characteristic of a two magnons scattering relaxation process with few inhomogeneities[29]. The value of this angle is sample dependent as it is related to the distribution of the magnetic inhomogeneities. The dominance in our films of those two

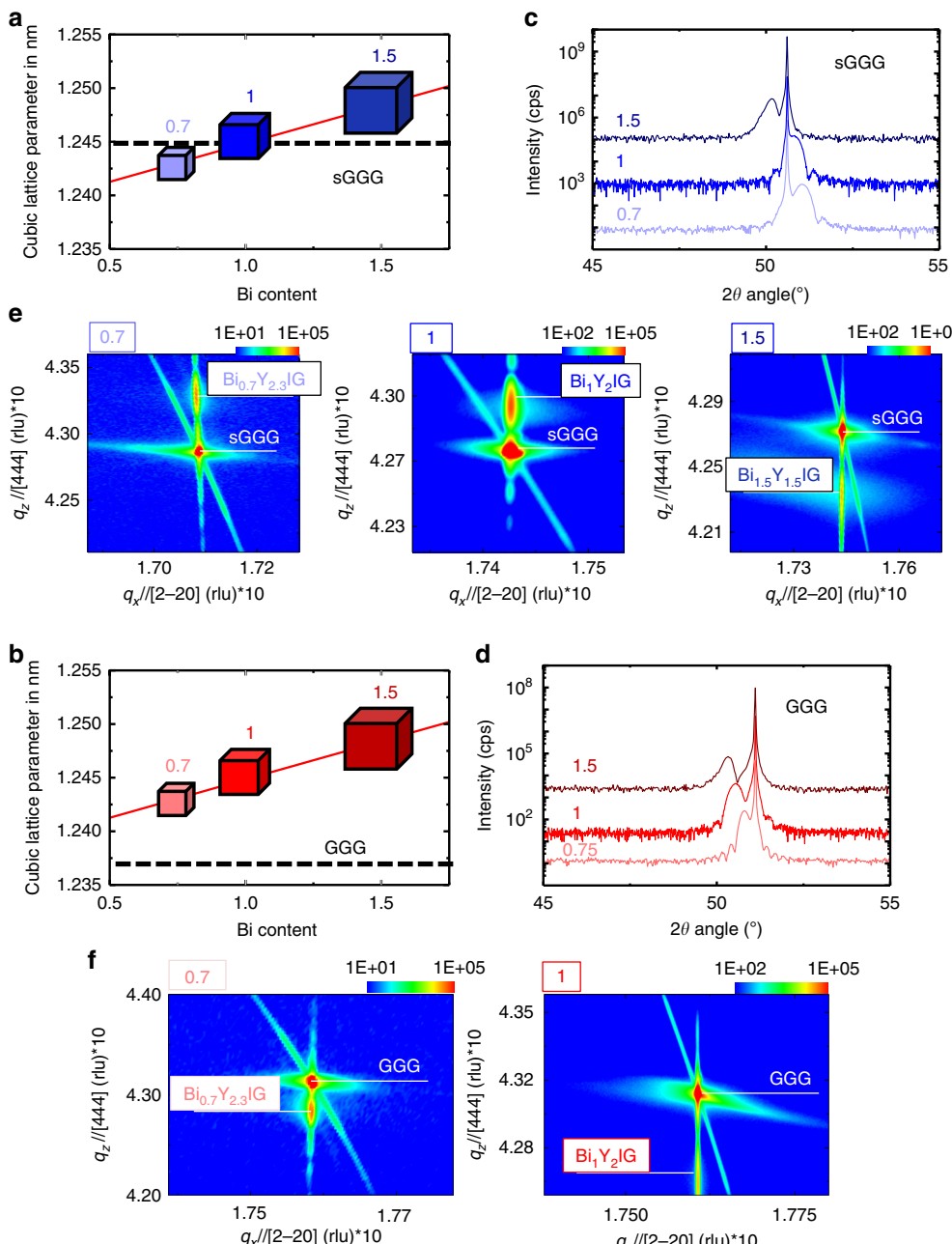

**Fig. 1** Structural properties of ultra-thin BiYIG films. **a** and **b** Evolution of the target cubic lattice parameter of $Bi_xY_{3-x}IG$, the dashed line represents the substrate (sGGG and GGG, respectively) lattice parameter and allows to infer the expected tensile or compressive strain arising for each substrate/target combination. **c** and **d** $2\theta-\omega$ X-Ray diffraction scan along the (111) out-of-plane direction for $Bi_xY_{3-x}IG$ films grown on sGGG (111) and GGG (111), respectively. From the film and substrate diffraction peak position, we can conclude about the nature of the strain. Compressive strain is observed for 1.5 doped films grown on sGGG substrate and for all films grown on GGG whereas tensile strain occurs for films with $x = 0.7$ and $x = 1$ Bi content grown on sGGG. **e** and **f** RSM along the evidence the (642) oblique plan showing pseudomorphic growth in films: both substrate and film the diffraction peak are aligned along the $q_x//[20-2]$ direction. The relative position of the diffraction peak of the film (up or down) along $q_x$ is related to the out-of-plane misfit between the substrate and the film (tensile or compressive)

**Table 1 Summary of the magnetic properties of Bi$_x$Y$_{3-x}$IG films on GGG and sGGG substrates**

| Bi doping | Substrate | $\mu_0 M_S$ (mT) | $\mu_0 M_{eff}$ (mT) | $\mu_0 H_{KU}$ (mT) |
|---|---|---|---|---|
| 0 | GGG | 157 | 200 | −43 |
| 0.7 | sGGG | 180 | −151 | 331 |
| 0.7 | GGG | 172 | 214 | −42 |
| 1 | sGGG | 172 | −29 | 201 |
| 1 | GGG | 160 | 189 | −29 |
| 1.5 | sGGG | 162 | 278 | −116 |

The saturation magnetization is roughly unchanged. The effective magnetization $M_{eff}$ obtained through broad-Band FMR measurements allow to deduce the out-of-plane anisotropy fields $H_{KU}$ ($H_{KU} = M_s - M_{eff}$) confirming the dramatic changes of the out-of-plane magnetic anisotropy variations observed in the hysteresis curves

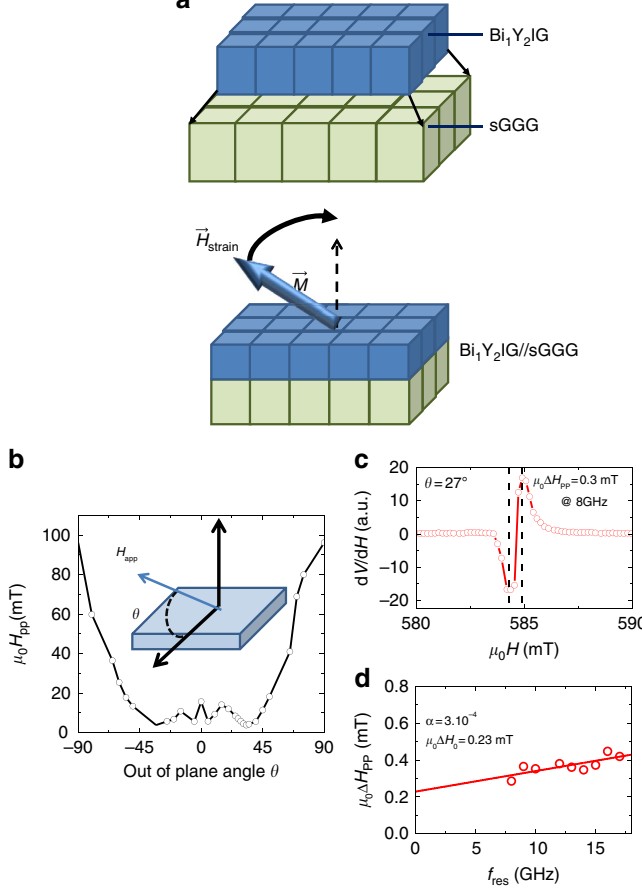

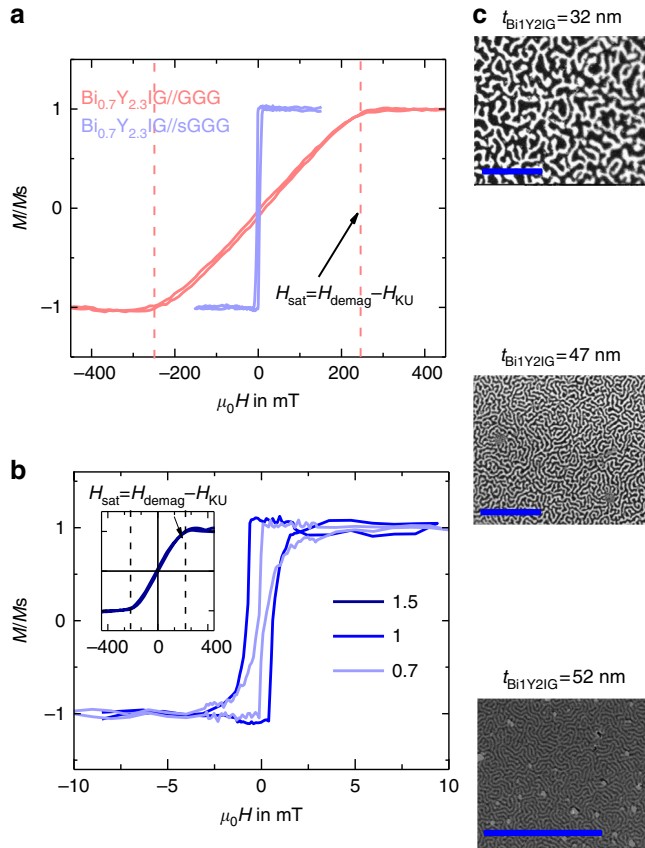

**Fig. 3** Dynamical properties of BiYIG films with PMA. **a** Sketch of the epitaxial configuration for Bi$_1$Y$_2$IG films, films are grown under tensile strain giving rise to tetragonal distortion of the unit cell. **b** Out-of-plane angular dependence of the peak-to-peak FMR linewidth ($\Delta H_{pp}$) at 8 GHz on a 30 nm thick Bi$_1$Y$_2$IG//sGGG with PMA (the continuous line is a guide for the eye). The geometry of the measurement is shown in top right of the graph. The wide disparity of the value for the peak to peak linewidth $\Delta H_{pp}$ is attributed to the two-magnons scattering process and inhomogeneities in the sample. **c** FMR absorption linewidth of 0.3 mT for the same film at measured at $\theta = 27°$. **d** Frequency dependence of the FMR linewidth. The calculated Gilbert damping parameter and the extrinsic linewidth are displayed on the graph

**Fig. 2** Static magnetic properties. **a** Out-of-plane Kerr hysteresis loop performed in the polar mode for Bi$_{0.7}$Y$_{2.3}$IG films grown on the two substrates: GGG and sGGG. **b** Same measurement for Bi$_x$Y$_{3-x}$IG grown on sGGG with the three different Bi doping ($x = 0.7$, 1, and 1.5). Bi$_{0.7}$Y$_{2.3}$IG//GGG is in-plane magnetized whereas perpendicular magnetic anisotropy (PMA) occurs for $x = 0.7$ and $x = 1$ films grown on sGGG: square shaped loops with low saturation field ($\mu_0 H_{sat}$ about 2.5 mT) are observed. Those two films are experiencing tensile strain. Whereas the inset shows that the Bi$_{1.5}$Y$_{1.5}$IG film saturates at a much higher field with a curve characteristic of in-plane easy magnetization direction. Note that for Bi$_{1.5}$Y$_{1.5}$IG//sGGG $\mu_0 H_{sat} \approx 290$ mT > $\mu_0 M_s \approx 162$ mT which points toward a negative uniaxial anisotropy term ($\mu_0 H_{KU}$) of 128 mT which is coherent with the values obtained from in-plane FMR measurement. **c** Magnetic domains structure imaged on Bi$_1$Y$_2$IG//sGGG films of three different thicknesses at remanant state after demagnetization. The scale bar, displayed in blue, equals 20 µm. Periods of the magnetic domains structure ($D_{width}$) are derived using 2D Fast Fourier Transform. We obtained $D_{width} = 3.1$, 1.6, and 0.4 µm for $t_{BiY2IG} = 32$, 47, and 52 nm, respectively. We note a decrease of $D_{width}$ with increasing $t_{BiY2IG}$ that is coherent with the Kaplan and Gehring model valide in the case $D_{width} \gg t_{BiYIG}$

intrinsic relaxation processes (Gilbert damping and two-magnons scattering) confirms the high films quality. We also derive the damping value of this film (Fig. 3d) by selecting the lowest linewidth (corresponding to a specific out-of-plane angle) at each frequency, the spread of the out-of-plane angle is ±3.5° around 30.5°. The obtained Gilbert damping value $\alpha = 3 \times 10^{-4}$ and the peak-to-peak extrinsic linewidth $\mu_0 \Delta H_0 = 0.23$ mT are comparable to the one obtained for the best PLD grown YIG//GGG nanometer thick films[28] ($\alpha = 2 \times 10^{-4}$). For $x = 0.7$ Bi doping, the smallest observed FMR linewidth is 0.5 mT at 8 GHz.

The low magnetic losses of BiYIG films could open new perspectives for magnetization dynamics control using spin-orbit torques[20,30,31]. For such phenomenon interface transparency to spin current is then the critical parameter which is defined using the effective spin-mixing conductance ($G_{\uparrow\downarrow}$). We use spin pumping experiments to estimate the increase of the Gilbert damping due to Pt deposition on Bi$_1$Y$_2$IG films. The spin-mixing conductance can thereafter be calculated using $G_{\uparrow\downarrow} = \frac{4\pi M_s t_{film}}{g_{eff}\mu_B}(\Delta\alpha)$

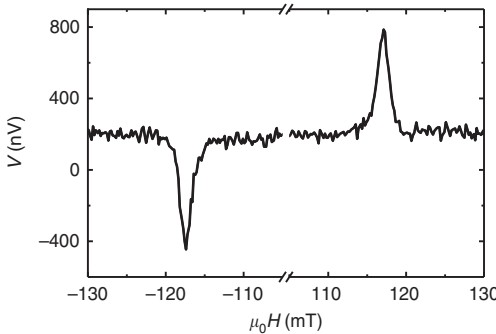

**Fig. 4** Inverse Spin Hall Effect of BiYIG films with in-plane magnetic anisotropy. Inverse Spin Hall Effect (ISHE) voltage vs magnetic field measured on the Pt/Bi$_{1.5}$Y$_{1.5}$IG//sGGG sample in the FMR resonant condition at 6 GHz proving the interface transparency to spin current. The rf excitation field is about 10$^{-3}$ mT which corresponds to a linear regime of excitation. Bi$_{1.5}$Y$_{1.5}$IG//sGGG presents an in-plane easy magnetization axis due to a growth under compressive strain

where $M_s$ and $t_{film}$ are the BiYIG magnetization and thickness, $g_{eff}$ is the effective Landé factor ($g_{eff} = 2$), $\mu_B$ is the Bohr magneton and $\Delta\alpha$ is the increase in the Gilbert damping constant induced by the Pt top layer. We obtain $G_{\uparrow\downarrow} = 3.9 \times 10^{18}\,\mathrm{m}^{-2}$ which is comparable to what is obtained on PLD grown YIG//GGG systems[28,32,33]. Consequently, the doping in Bi should not alter the spin orbit-torque efficiency and spin-torque devices made out of BiYIG will be as energy efficient as their YIG counterpart. To further confirm that spin current crosses the Pt/BiYIG interface, we measure Inverse Spin Hall Effect (ISHE) in Pt for a Pt/Bi$_{1.5}$Y$_{1.5}$IG(20 nm)//sGGG in-plane magnetized film (to fulfill the ISHE geometry requirements the magnetization needs to be in-plane and perpendicular to the measured voltage). We measure a characteristic voltage peak due to ISHE that reverses its sign when the static in-plane magnetic field is reversed (Fig. 4). We emphasize here that the amplitude of the signal is similar to that of Pt/YIG//GGG in the same experimental conditions.

## Conclusion

In summary, this new material platform will be highly beneficial for magnon-spintronics and related research fields like caloritronics. In many aspects, ultra-thin BiYIG films offer new leverages for fine tuning of the magnetic properties with no drawbacks compared to the reference materials of these fields: YIG. BiYIG with its higher Faraday rotation coefficient (almost two orders of magnitude more than that of YIG) will increase the sensitivity of light based detection techniques that can be used (Brillouin light spectroscopy (BLS) or time resolved Kerr microscopy[34]). Innovative schemes for on-chip magnon-light coupler could be now developed bridging the field of magnonics to the one of photonics. From a practical point of view, the design of future active devices will be much more flexible as it is possible to easily engineer the spin waves dispersion relation through magnetic anisotropy tuning without the need of large bias magnetic fields. For instance, working in the forward volume waves configuration comes now cost free, whereas in standard in-plane magnetized media one has to overcome the demagnetizing field. As the development of PMA tunnel junctions was key in developing today scalable MRAM technology, likewise, we believe that PMA in nanometer-thick low loss insulators paves the path to new approaches where the magnonic medium material could also be used to store information locally combining therefore the memory and computational functions, a most desirable feature for the brain-inspired neuromorphic paradigm.

## Methods

**Pulsed laser deposition (PLD) growth.** The PLD growth of BiYIG films is realized using stoichiometric BiYIG target. The laser used is a frequency tripled Nd:YAG laser ($\lambda = 355$ nm), of a 2.5 Hz repetition rate and a fluency varying from 0.95 to 1.43 J cm$^{-2}$ depending upon the Bi doping in the target. The distance between target and substrate is fixed at 44 mm. Prior to the deposition the substrate is annealed at 700 °C under 0.4 mbar of O$_2$. For the growth, the pressure is set at 0.25 mbar O$_2$ pressure. The optimum growth temperature varies with the Bi content from 400 to 550 °C. At the end of the growth, the sample is cooled down under 300 mbar of O$_2$.

**Structural characterization.** An Empyrean diffractometer with K$\alpha_1$ monochromator is used for measurement in Bragg-Brentano reflection mode to derive the (111) interatomic plan distance. Reciprocal Space Mapping is performed on the same diffractometer and we used the diffraction along the (642) plane direction which allow to gain information on the in-plane epitaxy relation along [20-2] direction.

**Magnetic characterization.** A quantum design SQUID magnetometer was used to measure the films' magnetic moment ($M_s$) by performing hysteresis curves along the easy magnetic direction at room temperature. The linear contribution of the paramagnetic (sGGG or GGG) substrate is linearly subtracted.

Kerr microscope (Evico Magnetics) is used in the polar mode to measure out-of-plane hysteresis curves at room temperature. The same microscope is also used to image the magnetic domains structure after a demagnetization procedure. The spatial resolution of the system is 300 nm.

A broadband FMR setup with a motorized rotation stage was used. Frequencies from 1 to 20 GHz have been explored. The FMR is measured as the derivative of microwave power absorption via a low frequency modulation of the DC magnetic field. Resonance spectra were recorded with the applied static magnetic field oriented in different geometries (in-plane or tilted of an angle $\theta$ out-of the stripline plane). For out-of-plane magnetized samples the Gilbert damping parameter has been obtained by studying the linewidth angular dependence. The procedure assumes that close to the minimum linewidth (Fig. 3a) most of the linewidth angular dependence is dominated by the inhomogeneous broadening, thus optimizing the angle for each frequency within few degrees allows to estimate better the intrinsic contribution. To do so we varied the out-of-plane angle of the static field from 27° to 34° for each frequency and we select the lowest value of $\Delta H_{pp}$.

For Inverse spin Hall effect measurements, the same FMR setup was used, however here the modulation is no longer applied to the magnetic field but to the RF power at a frequency of 5 kHz. A Stanford Research SR860 lock-in was used a signal demodulator.

**Data availability.** The data that support the findings of this study are available within the article or from the corresponding author upon reasonable request.

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

## Acknowledgements

We acknowledge J. Sampaio for preliminary Faraday rotation measurements and N. Reyren and A. Barthélémy for fruitful discussions. This research was supported by the ANR Grant ISOLYIG (ref 15-CE08-0030-01). L.S. is partially supported by G.I.E III-V Lab. France.

## Author contributions

L.S. performed the growth, all the measurements, the data analysis and wrote the manuscript with A.A., N.B. and J.B.Y. conducted the quantitative Faraday Rotation measurements and participated in the FMR data analysis. L.Q. and fabricated the PLD targets. R.L. supervised the target fabrication and participated in the design of the study. E.J. participated in the optimization of the film growth conditions. C.C. supervised the structural characterization experiments. A.A. conceived the study and was in charge of overall direction. P.B. and V.C. contributed to the design and implementation of the research. All authors discussed the results and commented on the manuscript.

## Additional information

**Competing interests:** The authors declare no competing interests.

