## [Peer Review File · Nature Communications]

Reviewers' comments:

Reviewer #1 (Remarks to the Author):

This work presents an important result for the magnonic community. The authors demonstrate a low damping PMA magnetic insulator BiYIG for magnonic device applications. PMA magnetic insulators allow dense integration of magnonic devices. They also allow using the forward volume waves for device applications. The challenge in previous PMA materials are the strong spin-orbit coupling ions induced magnon-phonon scattering and increased material damping. The authors used Bi doped YIG and the growth induced magnetic anisotropy to overcome this issue and demonstrate a very narrow linewidth PMA magnonic material. In general I believe this work is timely and important, which will be of general interest to the community. However there are several questions for the authors to respond before I can recommend publication of this work in Nature Communications.

1. Gilbert damping coefficient of the PMA material should be reported. As pointed out in the paper, this is the key performance factor of a magnonic insulator material. The authors should provide the frequency dependent FMR measurement results and the analysis of damping coefficient for their films, both PMA and in-plane anisotropic samples.
2. Please provide more insights on the structural origin of different growth induced magnetic anisotropy for BiYIG on different substrates. TEM or other structural analysis is preferred.
3. BiYIG thin films have been reported to show PMA in previous studies, especially in the case of very thin film thickness. The mechanism has been attributed to the growth induced anisotropy. From the underlying mechanism point of view, what is fundamentally new in this report? The authors need to provide more understanding on how to achieve low damping yet PMA material compared to previous reports.
4. The authors need to provide a more comprehensive review of previous works on PMA insulators for magnonic applications especially recent reports. For example Mn doped YIG thin films show inplane PMA down to 30 nm thickness with narrow FMR linewidths in a recent report. (Phys. Rev. B, 96, 224403, 2017)
5. Table S1, substrate instead of substrat. Please label Fig. S2 for the GGG and sGGG substrate cases.

Reviewer #2 (Remarks to the Author):

The paper is an experimental study of the development of Bi-substituted YIG films for spintronic studies. The authors grew high quality BiYIG films on GGG and SGGG. Varied strain by varying substrate (SGGG vs. GGG) and by varying the lattice parameter of the film itself by varying the content of the Bi dopant. The crystal structure characterization was carefully done and shows that films are certainly very high quality. These films show low Gilbert damping which is very important for spin transport and thus for spintronics. The achievement of PMA in low loss films is certainly useful and should generate a lot of interest to researchers in developing materials for insulator based spintronics.

However I do not believe that the study present substantial new fundamental insights into the material development or spintronic physics. As the authors point out, The fundamental idea of using strain for inducing PMA has received considerable attention recently: Li, P. et al. Nat. Commun. (2016) doi:10.1038/ncomms12688 which was REF 4 in the manuscript and C. Tang et al PRB 94, 140403(R), (2016) showed PMA in TIG films. The authors neglected to reference the latter. PMA caused by interfacial strain has also been demonstrated in the same Bi doped YIG system of the present study. This study should be reference in the present manuscript. Both the Li, P. et al. Nat. Commun. (2016) and C. Tang et al PRB not only developed new material approaches, but used the PMA in their films to help unravel some outstanding questions in spintronics. For example, the PRB work used varying film thickness and bi-and tri-layer structures to show that both the proximity-induced ferromagnetism and spin current contribute to the anomalous Hall effect. The Inverse Spin Hall Effect (ISHE) voltage vs magnetic field expedients in the present study show spin transparency. But the low damping + PMA is not used to probe any underlying phenomena. For these reasons I believe that these results are more suited to specialist in spintronics rather than the broader spintronic community.

Some minor suggestions:

- 1) It would be good to show both in plane and out of plane M-H measurements on the same sample to strengthen the PMA argument. This is difficult with Kerr measurements, but is easy with a Quantum design SQUID which is listed in characterization section.
- 2) Degrees.m⁻¹ are used as units for Faraday rotation. Since Faraday rotation depends linearly on magnetic field, the measured coefficients should reflect this, the preferred units are rad. T⁻¹. m⁻¹
- 3) The varying frequency FMR measurement used to extract the Gilbert damping coefficient should be shown.

Reviewer #3 (Remarks to the Author):

Recommendation: Accept with revisions.

This paper details a very interesting and important study, demonstrating perpendicular magnetic anisotropy in ultra thin garnet films. This is a very nice analysis via several different measurement techniques to demonstrate the control of anisotropy

Suggestions:

- 1) Abstract reads more like an introduction, it should state motivation more briefly and then summarize results. Also, it says "...none of them has PMA..." but in text it references a recent study with PMA, so "few of them have PMA" would be more accurate. Finally the abstract says that alpha (damping) is the main figure of merit, but then linewidth is given to show high quality. Best to state alpha and linewidth are important and then give the values of these since they are mentioned in the paper.
- 2) Introduction: Line 7 says "However, their large magnetic losses prohibit their use as spin-wave medium." It would be useful if authors would give numbers for these losses so that the reported values in this paper could be compared. Seems like Intro ends after "Gibert damping¹⁷." (line 11). A new paragraph could be started since results start to be presented.
- 3) The Faraday rotation is rather unbelievably high, but no details are given after this very short statement.
- 4) There is one statement about "growth induced anisotropy" which is a bit unsatisfying because the premise here is that the strain is used to control anisotropy. The argument in SI tries to detail the issue, but it is unsatisfying given the whole paper is about strain. Without further information, it seems best to leave out this speculation. Perhaps since strain is volumetric, increasing with thickness, TEM EDS could be done to see if Bi distribution changes along the thickness, which could be a strain effect.
- 5) Can the authors say more about HKu? What is M_{eff} that it is subtracted from M_s to get HKu and it also changes sign with in-plane to PMA.
- 6) On pg 3 "state of the art PLD grown in-plane magnetized YIG//GGG at "such resonant" conditions. Can authors provide details (how thick, what freq)? If there is a reference, also cite that.
- 7) Similarly, cite reference or give details about Pt/YIG//GGG
- 8) why was the SHE measured in an inplane film when the build up was all about PMA?

Figures:

- 1) Best to discuss "a-f" in order in the caption to step reader through the figure. "f" text labels are hard to read.
- 2) Add label "sGGG" to part "b". Maybe show all values of x on GGG as "a" instead of

repeating a sample. Why is Hsat so high? Seems larger than Hku would predict but it could be labeling of axis is just not clear. What is Hsat?

3) Part "a" is not discussed in caption at all. Seems like it is backward, unless the film is down (green) and substrate up (blue)? Maybe a label in figure would help. "d" and "e" seem opposite from text and caption descriptions

Excellent paper that could use some work, but should definitely be published with the suggested updates.

We are grateful for all three referees for their careful reading of the paper. Their questions and comments allowed us to improve the quality and the clarity of our paper. We are addressing below each point that they have raised and hope that they will find our answers satisfactory.

Reviewer #1 (Remarks to the Author):

This work presents an important result for the magnonic community. The authors demonstrate a low damping PMA magnetic insulator BiYIG for magnonic device applications. PMA magnetic insulators allow dense integration of magnonic devices. They also allow using the forward volume waves for device applications. The challenge in previous PMA materials are the strong spin-orbit coupling ions induced magnon-phonon scattering and increased material damping. The authors used Bi doped YIG and the growth induced magnetic anisotropy to overcome this issue and demonstrate a very narrow linewidth PMA magnonic material. In general I believe this work is timely and important, which will be of general interest to the community. However there are several questions for the authors to respond before I can recommend publication of this work in Nature Communications.

We thank the referee for his/her positive evaluation and for pointing out the relevance of this work and also to highlight that it can be of a general interest for the community.

1. Gilbert damping coefficient of the PMA material should be reported. As pointed out in the paper, this is the key performance factor of a magnonic insulator material. The authors should provide the frequency dependent FMR measurement results and the analysis of damping coefficient for their films, both PMA and in-plane anisotropic samples.

We agree with the referee that this was a missing information. We have updated the data and changed Figure 3. Figure 3-c shows now the frequency dependence of the FMR linewidth. The best results we obtained for a PMA film ($\text{Bi}_1\text{Y}_2\text{IG}$) is a damping of $3 \cdot 10^{-4}$ with an extrinsic linewidth of 0.23 mT. We have also added in the supplementary materials the FMR characteristics of an in-plane anisotropy $\text{Bi}_{1.5}\text{Y}_{1.5}\text{IG}$ film.

2. Please provide more insights on the structural origin of different growth induced magnetic anisotropy for BiYIG on different substrates. TEM or other structural analysis is preferred.

We have indeed tried to perform TEM studies, however our attempts are for the moment unsuccessful. We stress however that TEM and related technics like STEM-EELS spectroscopies even if they are the most powerful tools available to characterize local structures of ultra-thin films may not be effective in our case. The large number of dodecahedral sites in a single unit cell (8) will probably not allow to discriminate preferential substitution sites at the origin of the growth induced anisotropy. Nevertheless, we are in the process of conducting those studies but this will take over a year. It is our objective to conduct deeper studies to clarify the physical origin of the observed growth induced anisotropy.

3. BiYIG thin films have been reported to show PMA in previous studies, especially in the case of very thin film thickness. The mechanism has been attributed to the growth induced anisotropy. From the

underlying mechanism point of view, what is fundamentally new in this report? The authors need to provide more understanding on how to achieve low damping yet PMA material compared to previous reports.

We do agree that the novelty of our report does not lie in the PMA alone neither the low damping alone but in the ability to conciliate those two properties that are usually mutually exclusive. It is probable that our findings are firstly due to the high epitaxial quality of our films that reduces extrinsic sources of magnetic relaxation. Moreover, there is no equivalent report where such epitaxial quality has been combined with relatively low Bi doping levels. As a matter of fact, most of the studies of BiYIG were focused on the achievement of very high Faraday rotation coefficient for photonic application, the objective was to increase the Bi content as much as possible which end-up to be detrimental to the dynamical properties. For example, in our case for $x=1.5$ Bi content the Gilbert damping is of $2 \cdot 10^{-3}$ even if the film is in-plane magnetized so too high Bi doping seems to affect the dynamic quality of the film. This information was missing in our initial submission, it is now in the supplementary information section.

4. The authors need to provide a more comprehensive review of previous works on PMA insulators for magnonic applications especially recent reports. For example Mn doped YIG tin films show inplane PMA down to 30 nm thickness with narrow FMR linewidths in a recent report. (Phys. Rev. B, 96, 224403, 2017)

We have complied with the referee suggestion and added the 2 references (Phys. Rev. B, 96, 224403, 2017 and *Appl. Phys. Lett.* **110**, 202403, 2017) that address PMA. However none of those reports shows a spontaneous out-of-plane magnetisation, the PMA was not large enough. The first report is the only one with dynamical data and the linewidth corresponding to the highest PMA value is 6.5 mT which is about 20 times larger than what we observe. We also have added an other reference following reviewer #2 suggestion of PMA in TIG but then the reported linewidth is 16.7 mT, 50 times larger than our findings.

5. Table S1, substrate instead of substrat. Please label Fig. S2 for the GGG and sGGG substrate cases.

We have corrected this error.

Reviewer #2 (Remarks to the Author):

The paper is an experimental study of the development of Bi-substituted YIG films for spintronic studies. The authors grew high quality BiYIG films on GGG and SGGG. Varied strain by varying substrate (SGGG vs. GGG) and by varying the lattice parameter of the film itself by varying the content of the Bi dopant. The crystal structure characterization was carefully done and shows that films are certainly very high quality. These films show low Gilbert damping which is very important for spin transport and thus for spintronics. The achievement of PMA in low loss films is certainly useful and should generate a lot of interest to researchers in developing materials for insulator based spintronics.

We thank the referee for recognizing the broad impact of our study for the spintronics community. We are confident that even more interest will come from growing communities of insulating magnonics and

caloritronics which are until now very much restricted to the use of YIG films with in-plane magnetization since no alternative material allows for long enough spin wave propagation length. We stress that the Gilbert damping (or the FMR linewidth) acts exponentially on the SW propagation length. Our report, in comparison to published results of PMA materials, is a factor of 50 improvement of the damping which leads to about 10^{22} improvement in the SW propagation length and therefore about 10^{22} improvement of magnonic devices operation made out-of BiYIG over any PMA material available.

However I do not believe that the study present substantial new fundamental insights into the material development or spintronic physics. As the authors point out, The fundamental idea of using strain for inducing PMA has received considerable attention recently: Li, P. et al. Nat. Commun. (2016) doi:10.1038/ncomms12688 which was REF 4 in the manuscript and C. Tang et al PRB 94, 140403(R), (2016) showed PMA in TIG films. The authors neglected to reference the latter. PMA caused by interfacial strain has also been demonstrated in the same Bi doped YIG system of the present study. This study should be reference in the present manuscript. Both the Li, P. et al. Nat. Commun. (2016) and C. Tang et al PRB not only developed new material approaches, but used the PMA in their films to help unravel some outstanding questions in spintronics.

As pointed out by the referee we did explicitly address the TIG system in the text "*Tm₃Fe₅O₁₂ or BaFe₁₂O₁₉ (respectively a garnet and an hexaferrite) have been used to demonstrate spin-orbit-torque magnetization reversal using a Pt over-layer as a source of spin current. However, their large magnetic losses prohibit their use as a spin-wave medium*". In this version we have corrected our mistake of not citing C. Tang et al.. Actually this citation is necessary as they are the only group that has measured the damping of TIG thin films, they did found a very decent (and expected) value of the linewidth (16.7 mT). A value that is 50 times larger than our findings. This makes our point: rather than being just an incremental improvement from previous work done on PMA with magnetic insulators our findings concerning the high dynamical quality of our films is opening unprecedented perspectives for the fields of magnonics, caloritronics and spin-orbitronics.

For example, the PRB work used varying film thickness and bi-and tri-layer structures to show that both the proximity-induced ferromagnetism and spin current contribute to the anomalous Hall effect. The Inverse Spin Hall Effect (ISHE) voltage vs magnetic field expedients in the present study show spin transparency. But the low damping + PMA is not used to probe any underlying phenomena.

The referee is right that the focus of our report is to show that low damping and PMA are compatible. We believe that this by itself is an achievement. We agree that the next step is to use this new material platform as a probing tool for underling physical phenomena. Those researches are actually under progress exploring different directions with many collaborations. Proximity effect is one of those physical phenomena, others are about THz emission, magnonics crystals physics, coupling to photonics thanks the very large magneto-optical coefficient. This however goes beyond the scope of the present publication.

For these reasons I believe that these results are more suited to specialist in spintronics rather than the broader spintronic community.

We respectfully disagree with this comment as the material platform that we propose is relevant to other topics that make use of magnetic materials, spintronics, magnonics, THz, topological insulators ...

Some minor suggestions:

1) It would be good to show both in plane and out of plane M-H measurements on the same sample to strengthen the PMA argument. This is difficult with Kerr measurements, but is easy with a Quantum design SQUID which is listed in characterization section.

We believe that showing domains configuration with polar MOKE is enough to prove our points as we present raw data. Given the high paramagnetic contribution of the sGGG substrate M-H measurements based on SQUID along the hard axis relies on careful background subtraction which could be subject to questioning. This is not the case with MOKE images.

2) Degrees.m⁻¹ are used as units for Faraday rotation. Since Faraday rotation depends linearly on magnetic field, the measured coefficients should reflect this, the preferred units are rad. T⁻¹. m⁻¹.

We respectfully disagree: expressing the Faraday rotation in deg. T⁻¹. m⁻¹ makes sense for paramagnetic or diamagnetic materials. However for ferromagnets and ferrimagnets the Faraday rotation is proportional to the magnetic induction B ($B=\mu_0(H+M)$). M of course dominates the response and as it is highly nonlinear with H (this our case here as the remanence is close to 100%), Faraday rotation for magnetic materials is the value obtained at magnetic saturation. It is usually expressed in deg. m⁻¹.

3) The varying frequency FMR measurement used to extract the Gilbert damping coefficient should be shown.

We do agree with this point. We have updated the data. We have changed Figure 3 that now shows in panel c the frequency dependence of the FMR linewidth and an example of a typical absorption line. The best results we obtained for a PMA film (Bi₁Y₂IG) is a damping of $3 \cdot 10^{-4}$ with an extrinsic linewidth of 0.23 mT.

Reviewer #3 (Remarks to the Author):

Recommendation: Accept with revisions.

This paper details a very interesting and important study, demonstrating perpendicular magnetic anisotropy in ultra thin garnet films. This is a very nice analysis via several different measurement techniques to demonstrate the control of anisotropy

We thank the referee for her/his positive evaluation

Suggestions:

1) Abstract reads more like an introduction, it should state motivation more briefly and then summarize results.

We have complied with referee request and changed the layout of our paper to fulfill those of Nature Communication Journal.

Also, it says "...none of them has PMA..." but in text it references a recent study with PMA, so "few of them have PMA" would be more accurate.

We have clarified this point. Our full sentence reads now as "*This makes the number of relevant materials for SW propagation quite limited and none of them has yet been found to possess a large enough perpendicular magnetic anisotropy (PMA) to induce spontaneous out-of-plane magnetization.*". We agree on the fact that PMA magnetic insulators have already been achieved (TIG for example) but what we want to stress is that the combination of low damping values and PMA have so far not been achieved. In this sentence we are referring to "*materials relevant for spin wave propagation*" and so far no material with PMA and damping values below 10^{-3} have yet been achieved.

Finally the abstract says that alpha (damping) is the main figure of merit, but then linewidth is given to show high quality. Best to state alpha and linewidth are important and then give the values of these since they are mentioned in the paper.

We fully agree with this comment. We have updated the data. We have changed Figure 3 that now shows in panel c the frequency dependence of the FMR linewidth and an example of a typical absorption line. The best results we obtained for a PMA film ($\text{Bi}_1\text{Y}_2\text{IG}$) is a damping of $3 \cdot 10^{-4}$ with an extrinsic linewidth of 0.23 mT.

2) Introduction: Line 7 says "However, their large magnetic losses prohibit their use as spin-wave medium." It would be useful if authors would give numbers for these losses so that the reported values in this paper could be compared.

We thank the referee for her/his suggestion : this sentence reads now as : "*However, their large magnetic losses prohibit their use as a spin-wave medium (reported value of $\mu_0\Delta H_{pp}$ of TIG is 16.7 mT at 9.5 GHz) Tang et al. 2017*".

Seems like Intro ends after "Gibert damping17." (line 11). A new paragraph could be started since results start to be presented.

We have complied with referee's suggestion.

3) The Faraday rotation is rather unbelievably high, but no details are given after this very short statement.

The referee highlights an interesting point concerning our results, first it is worth to notice that the value is not so high compared to other reports and the value that we mentioned is the highest obtained among all our films.

Reports on the Faraday rotation coefficient of BiYIG films using PLD growth technique showed an increase of $\theta_F = -1.9^\circ/\mu\text{m}$ @ 630 nm per Bi atoms (Chern et al.). However, the film thickness plays also a role in the value of the Faraday rotation coefficient (Kahl et al.) and this latest point is not well understood. The value reported in the paper is the one obtained on the thinnest films of 15 nm with 1 Bi doping. We have clarified this in the present version. We have extended the discussion on the Faraday and added 2 references.

4) There is one statement about “growth induced anisotropy” which is a bit unsatisfying because the premise here is that the strain is used to control anisotropy. The argument in SI tries to detail the issue, but it is unsatisfying given the whole paper is about strain. Without further information, it seems best to leave out this speculation.

We fully agree with the referee that we do not control the growth induced anisotropy. However, this anisotropy term is necessary to explain the large PMA. It has been modeled using the Van Vleck pair ordering theory (Callen et al 1971, DOI: [http://doi.org/10.1016/0025-5408\(71\)90071-7](http://doi.org/10.1016/0025-5408(71)90071-7)). We estimate the magneto-elastic term to be about 25 to 30% of the total PMA, rather than living out this discussion completely we chose to present modeling of the large PMA in the supplementary information section for more expert readers. We hope that this argument is convincing enough to the referee.

Perhaps since strain is volumetric, increasing with thickness, TEM EDS could be done to see if Bi distribution changes along the thickness, which could be a strain effect.

We have indeed tried to perform TEM studies, however our attempts are for the moment unsuccessful. We stress however that TEM and related technics like STEM-EELS spectroscopies even if they are the most powerful tools available to characterize local structures of ultra-thin films may not be effective in our case. The large number of dodecahedral sites in a single unit cell (8) will probably not allow to discriminate preferential substitution sites at the origin of the growth induced anisotropy. Nevertheless, we are in the process of conducting those studies but this will take over a year. It is our objective to clarify the physical origin of the observed growth anisotropy, however answering this question right now is impossible.

5) Can the authors say more about HKu? What is M_{eff} that it is subtracted from M_s to get HKu and it also changes sign with in-plane to PMA.

We agree that this information was missing. We use the regular Kittel formula to extract M_{eff} from the in-plane FMR measurements (resonance frequency vs. magnetic field) then we use the fact that the effective magnetization M_{eff} is the sum of the saturation magnetization and the uniaxial anisotropy field $H_{\text{Ku}} = M_s - M_{\text{eff}}$. We have clarified this procedure in the supplementary information.

6) On pg 3 “state of the art PLD grown in-plane magnetized YIG//GGG at “such resonant” conditions. Can authors provide details (how thick, what freq)? If there is a reference, also cite that.

We have added a reference for YIG films grown using PLD with the same thickness that show a Gilbert damping parameter of $2.3 \cdot 10^{-4}$ (d'Allivy Kelly et al. APL 2013). This report is a self-citation.

7) Similarly, cite reference or give details about Pt/YIG//GGG

This point has also been addressed by d'Allivy Kelly et al. APL 2013. We can therefore guaranty that the experimental procedure is equivalent.

8) why was the SHE measured in an inplane film when the build up was all about PMA?

The FMR induced inverse spin Hall measurements is only sensitive to the in-plane component of the magnetization as the voltage is the result of a cross product:

$$\overrightarrow{V}_{ISHE} \propto \overrightarrow{J}_S \times \overrightarrow{M}$$

Therefore, we used a BiYIG film in-plane magnetized to perform the ISHE measurement and qualify the interface for spin transparency.

Figures:

- 1) Best to discuss "a-f" in order in the caption to step reader through the figure. "f" text labels are hard to read.

We have changed the labeling and modified the figure caption to make it more easy to read.

- 2) Add label "sGGG" to part "b". Maybe show all values of x on GGG as "a" instead of repeating a sample. Why is Hsat so high? Seems larger than Hku would predict but it could be labeling of axis is just not clear. What is Hsat?

We thank the referee for its careful reading of the paper. This measurement was performed on a sample coming from a different growth batch than the one used for FMR. We have now corrected this discrepancy. We have redone the measurement on the same sample used in the FMR data.

3) Part "a" is not discussed in caption at all. Seems like it is backward, unless the film is down (green) and substrate up (blue)? Maybe a label in figure would help. "d" and "e" seem opposite from text and caption descriptions

We thank the referee for its careful reading of the paper. We have taking into account his suggestion and corrected our mislabeling.

Excellent paper that could use some work, but should definitely be published with the suggested updates.

We sincerely thank the referee for his help and his positive evaluation

Reviewers' Comments:

Reviewer #1 (Remarks to the Author):

The authors reported the very low Gilbert damping coefficient of this PMA film, which is novel enough now to be considered publication. I can now recommend publication of this paper. I still suggest the authors to provide more structural information of the PMA origin, if possible in their final form of the submission.

Reviewer #2 (Remarks to the Author):

The manuscript has improved significantly and my concerns have been addressed. I do recommend adding the following reference since it shows work on strain tuning PMA in the same BiYIG system that is being addressed here:

P. Sellappan et al “An integrated approach to doped thin films with strain tunable magnetic anisotropy: Powder synthesis, target preparation and pulsed laser deposition of Bi:YIG”
Materials Research Letters, (2017) 5, 41-47.
DOI: 10.1080/21663831.2016.1195779.

Reviewer #3 (Remarks to the Author):

This re-write is sufficient for publication. Accept as submitted.

Response to referees

First reply of referees: 28/03/2018

We are grateful for all three referees for their careful reading of the paper. Their questions and comments allowed us to improve the quality and the clarity of our paper. We are addressing below each point that they have raised and hope that they will find our answers satisfactory.

Reviewer #1 (Remarks to the Author):

This work presents an important result for the magnonic community. The authors demonstrate a low damping PMA magnetic insulator BiYIG for magnonic device applications. PMA magnetic insulators allow dense integration of magnonic devices. They also allow using the forward volume waves for device applications. The challenge in previous PMA materials are the strong spin-orbit coupling ions induced magnon-phonon scattering and increased material damping. The authors used Bi doped YIG and the growth induced magnetic anisotropy to overcome this issue and demonstrate a very narrow linewidth PMA magnonic material. In general I believe this work is timely and important, which will be of general interest to the community. However there are several questions for the authors to respond before I can recommend publication of this work in Nature Communications.

We thank the referee for his/her positive evaluation and for pointing out the relevance of this work and also to highlight that it can be of a general interest for the community.

1. Gilbert damping coefficient of the PMA material should be reported. As pointed out in the paper, this is the key performance factor of a magnonic insulator material. The authors should provide the frequency dependent FMR measurement results and the analysis of damping coefficient for their films, both PMA and in-plane anisotropic samples.

We agree with the referee that this was a missing information. We have updated the data and changed Figure 3. Figure 3-c shows now the frequency dependence of the FMR linewidth. The best results we obtained for a PMA film (Bi₁Y₂IG) is a damping of $3 \cdot 10^{-4}$ with an extrinsic linewidth of 0.23 mT. We have also added in the supplementary materials the FMR characteristics of an in-plane anisotropy Bi_{1.5}Y_{1.5}IG film.

2. Please provide more insights on the structural origin of different growth induced magnetic anisotropy for BiYIG on different substrates. TEM or other structural analysis is preferred.

We have indeed tried to perform TEM studies, however our attempts are for the moment unsuccessful. We stress however that TEM and related technics like STEM-EELS spectroscopies even if they are the most powerful tools available to characterize local structures of ultra-thin films may not be effective in our case. The large number of dodecahedral sites in a single unit cell (8) will probably not allow to

discriminate preferential substitution sites at the origin of the growth induced anisotropy. Nevertheless, we are in the process of conducting those studies but this will take over a year. It is our objective to conduct deeper studies to clarify the physical origin of the observed growth induced anisotropy.

3. BiYIG thin films have been reported to show PMA in previous studies, especially in the case of very thin film thickness. The mechanism has been attributed to the growth induced anisotropy. From the underlying mechanism point of view, what is fundamentally new in this report? The authors need to provide more understanding on how to achieve low damping yet PMA material compared to previous reports.

We do agree that the novelty of our report does not lie in the PMA alone neither the low damping alone but in the ability to conciliate those two properties that are usually mutually exclusive. It is probable that our findings are firstly due to the high epitaxial quality of our films that reduces extrinsic sources of magnetic relaxation. Moreover, there is no equivalent report where such epitaxial quality has been combined with relatively low Bi doping levels. As a matter of fact, most of the studies of BiYIG were focused on the achievement of very high Faraday rotation coefficient for photonic application, the objective was to increase the Bi content as much as possible which end-up to be detrimental to the dynamical properties. For example, in our case for $x=1.5$ Bi content the Gilbert damping is of $2 \cdot 10^{-3}$ even if the film is in-plane magnetized so too high Bi doping seems to affect the dynamic quality of the film. This information was missing in our initial submission, it is now in the supplementary information section.

4. The authors need to provide a more comprehensive review of previous works on PMA insulators for magnonic applications especially recent reports. For example Mn doped YIG tin films show inplane PMA down to 30 nm thickness with narrow FMR linewidths in a recent report. (Phys. Rev. B, 96, 224403, 2017)

We have complied with the referee suggestion and added the 2 references (Phys. Rev. B, 96, 224403, 2017 and *Appl. Phys. Lett.* **110**, 202403, 2017) that adress PMA. However none of those reports shows a spontaneous out-of-plane magnetisation, the PMA was not large enough. The first report is the only one with dynamical data and the linewidth corresponding to the highest PMA value is 6.5 mT wich is about 20 times larger than what we observe. We also have added an other reference following reviewer #2 suggestion of PMA in TIG but then the reported linewidth is 16.7 mT, 50 times larger than our findings.

5. Table S1, substrate instead of substrat. Please label Fig. S2 for the GGG and sGGG substrate cases.

We have corrected this error.

Reviewer #2 (Remarks to the Author):

The paper is an experimental study of the development of Bi-substituted YIG films for spintronic studies. The authors grew high quality BiYIG films on GGG and SGGG. Varied strain by varying substrate (SGGG vs. GGG) and by varying the lattice parameter of the film itself by varying the content of the Bi dopant. The crystal structure characterization was carefully done and shows that films are certainly very high quality.

These films show low Gilbert damping which is very important for spin transport and thus for spintronics. The achievement of PMA in low loss films is certainly useful and should generate a lot of interest to researchers in developing materials for insulator based spintronics.

We thank the referee for recognizing the broad impact of our study for the spintronics community. We are confident that even more interest will come from growing communities of insulating magnonics and caloritronics which are until now very much restricted to the use of YIG films with in-plane magnetization since no alternative material allows for long enough spin wave propagation length. We stress that the Gilbert damping (or the FMR linewidth) acts exponentially on the SW propagation length. Our report, in comparison to published results of PMA materials, is a factor of 50 improvement of the damping which leads to about 10^{22} improvement in the SW propagation length and therefore about 10^{22} improvement of magnonic devices operation made out-of BiYIG over any PMA material available.

However I do not believe that the study present substantial new fundamental insights into the material development or spintronic physics. As the authors point out, The fundamental idea of using strain for inducing PMA has received considerable attention recently: Li, P. et al. Nat. Commun. (2016) doi:10.1038/ncomms12688 which was REF 4 in the manuscript and C. Tang et al PRB 94, 140403(R), (2016) showed PMA in TIG films. The authors neglected to reference the latter. PMA caused by interfacial strain has also been demonstrated in the same Bi doped YIG system of the present study. This study should be reference in the present manuscript. Both the Li, P. et al. Nat. Commun. (2016) and C. Tang et al PRB not only developed new material approaches, but used the PMA in their films to help unravel some outstanding questions in spintronics.

As pointed out by the referee we did explicitly address the TIG system in the text "*Tm₃Fe₅O₁₂ or BaFe₁₂O₁₉ (respectively a garnet and an hexaferrite) have been used to demonstrate spin-orbit-torque magnetization reversal using a Pt over-layer as a source of spin current. However, their large magnetic losses prohibit their use as a spin-wave medium*". In this version we have corrected our mistake of not citing C. Tang et al.. Actually this citation is necessary as they are the only group that has measured the damping of TIG thin films, they did found a very decent (and expected) value of the linewidth (16.7 mT). A value that is 50 times larger than our findings. This makes our point: rather than being just an incremental improvement from previous work done on PMA with magnetic insulators our findings concerning the high dynamical quality of our films is opening unprecedented perspectives for the fields of magnonics, caloritronics and spin-orbitronics.

For example, the PRB work used varying film thickness and bi-and tri-layer structures to show that both the proximity-induced ferromagnetism and spin current contribute to the anomalous Hall effect. The Inverse Spin Hall Effect (ISHE) voltage vs magnetic field expedients in the present study show spin transparency. But the low damping + PMA is not used to probe any underlying phenomena. The referee is right that the focus of our report is to show that low damping and PMA are compatible. We believe that this by itself is an achievement. We agree that the next step is to use this new material platform as a probing tool for underling physical phenomena. Those researches are actually under progress exploring different directions with many collaborations. Proximity effect is one of those physical phenomena, others are about THz emission, magnonics crystals physics, coupling to photonics thanks

the very large magneto-optical coefficient. This however goes beyond the scope of the present publication.

For these reasons I believe that these results are more suited to specialist in spintronics rather than the broader spintronic community.

We respectfully disagree with this comment as the material platform that we propose is relevant to other topics that make use of magnetic materials, spintronics, magnonics, THz, topological insulators ...

Some minor suggestions:

1) It would be good to show both in plane and out of plane M-H measurements on the same sample to strengthen the PMA argument. This is difficult with Kerr measurements, but is easy with a Quantum design SQUID which is listed in characterization section.

We believe that showing domains configuration with polar MOKE is enough to prove our points as we present raw data. Given the high paramagnetic contribution of the sGGG substrate M-H measurements based on SQUID along the hard axis relies on careful background subtraction which could be subject to questioning. This is not the case with MOKE images.

2) Degrees.m⁻¹ are used as units for Faraday rotation. Since Faraday rotation depends linearly on magnetic field, the measured coefficients should reflect this, the preferred units are rad. T⁻¹. m⁻¹.

We respectfully disagree: expressing the Faraday rotation in deg. T⁻¹. m⁻¹ makes sense for paramagnetic or diamagnetic materials. However for ferromagnets and ferrimagnets the Faraday rotation is proportional to the magnetic induction B ($B=\mu_0(H+M)$). M of course dominates the response and as it is highly nonlinear with H (this our case here as the remanence is close to 100%), Faraday rotation for magnetic materials is the value obtained at magnetic saturation. It is usually expressed in deg. m⁻¹.

3) The varying frequency FMR measurement used to extract the Gilbert damping coefficient should be shown.

We do agree with this point. We have updated the data. We have changed Figure 3 that now shows in panel c the frequency dependence of the FMR linewidth and an example of a typical absorption line. The best results we obtained for a PMA film (Bi₁Y₂IG) is a damping of $3 \cdot 10^{-4}$ with an extrinsic linewidth of 0.23 mT.

Reviewer #3 (Remarks to the Author):

Recommendation: Accept with revisions.

This paper details a very interesting and important study, demonstrating perpendicular magnetic anisotropy in ultra thin garnet films. This is a very nice analysis via several different measurement techniques to demonstrate the control of anisotropy

We thank the referee for her/his positive evaluation

Suggestions:

1) Abstract reads more like an introduction, it should state motivation more briefly and then summarize results.

We have complied with referee request and changed the layout of our paper to fulfill those of Nature Communication Journal.

Also, it says "...none of them has PMA..." but in text it references a recent study with PMA, so "few of them have PMA" would be more accurate.

We have clarified this point. Our full sentence reads now as "*This makes the number of relevant materials for SW propagation quite limited and none of them has yet been found to possess a large enough perpendicular magnetic anisotropy (PMA) to induce spontaneous out-of-plane magnetization.*". We agree on the fact that PMA magnetic insulators have already been achieved (TIG for example) but what we want to stress is that the combination of low damping values and PMA have so far not been achieved. In this sentence we are referring to "*materials relevant for spin wave propagation*" and so far no material with PMA and damping values below 10^{-3} have yet been achieved.

Finally the abstract says that alpha (damping) is the main figure of merit, but then linewidth is given to show high quality. Best to state alpha and linewidth are important and then give the values of these since they are mentioned in the paper.

We fully agree with this comment. We have updated the data. We have changed Figure 3 that now shows in panel c the frequency dependence of the FMR linewidth and an example of a typical absorption line. The best results we obtained for a PMA film ($\text{Bi}_1\text{Y}_2\text{IG}$) is a damping of $3 \cdot 10^{-4}$ with an extrinsic linewidth of 0.23 mT.

2) Introduction: Line 7 says "However, their large magnetic losses prohibit their use as spin-wave medium." It would be useful if authors would give numbers for these losses so that the reported values in this paper could be compared.

We thank the referee for her/his suggestion : this sentence reads now as : "*However, their large magnetic losses prohibit their use as a spin-wave medium (reported value of $\mu_0\Delta H_{pp}$ of TIG is 16.7 mT at 9.5 GHz) Tang et al. 2017*".

Seems like Intro ends after "Gibert damping17." (line 11). A new paragraph could be started since results start to be presented.

We have complied with referee's suggestion.

3) The Faraday rotation is rather unbelievably high, but no details are given after this very short statement.

The referee highlights an interesting point concerning our results, first it is worth to notice that the value is not so high compared to other reports and the value that we mentioned is the highest obtained among all our films. Reports on the Faraday rotation coefficient of BiYIG films using PLD growth technique showed an increase of $\theta_F = -1.9^\circ/\mu\text{m}$ @ 630 nm per Bi atoms (Chern et al.). However, the film thickness plays also a role in the value of the Faraday rotation coefficient (Kahl et al.) and this latest point is not well understood. The value reported in the paper is the one obtained on the thinnest films of 15 nm with 1 Bi doping. We have clarified this in the present version. We have extended the discussion on the Faraday and added 2 references.

4) There is one statement about "growth induced anisotropy" which is a bit unsatisfying because the premise here is that the strain is used to control anisotropy. The argument in SI tries to detail the issue, but it is unsatisfying given the whole paper is about strain. Without further information, it seems best to leave out this speculation.

We fully agree with the referee that we do not control the growth induced anisotropy. However, this anisotropy term is necessary to explain the large PMA. It has been modeled using the Van Vleck pair ordering theory (Callen et al 1971, DOI: [http://doi.org/10.1016/0025-5408\(71\)90071-7](http://doi.org/10.1016/0025-5408(71)90071-7)). We estimate the magneto-elastic term to be about 25 to 30% of the total PMA, rather than living out this discussion completely we chose to present modeling of the large PMA in the supplementary information section for more expert readers. We hope that this argument is convincing enough to the referee.

Perhaps since strain is volumetric, increasing with thickness, TEM EDS could be done to see if Bi distribution changes along the thickness, which could be a strain effect.

We have indeed tried to perform TEM studies, however our attempts are for the moment unsuccessful. We stress however that TEM and related technics like STEM-EELS spectroscopies even if they are the most powerful tools available to characterize local structures of ultra-thin films may not be effective in our case. The large number of dodecahedral sites in a single unit cell (8) will probably not allow to discriminate preferential substitution sites at the origin of the growth induced anisotropy. Nevertheless, we are in the process of conducting those studies but this will take over a year. It is our objective to clarify the physical origin of the observed growth anisotropy, however answering this question right now is impossible.

5) Can the authors say more about HKu? What is M_{eff} that it is subtracted from M_s to get HKu and it also changes sign with in-plane to PMA.

We agree that this information was missing. We use the regular Kittel formula to extract M_{eff} from the in-plane FMR measurements (resonance frequency vs. magnetic field) then we use the fact that the

effective magnetization M_{eff} is the sum of the saturation magnetization and the uniaxial anisotropy field $H_{\text{KU}} = M_{\text{S}} - M_{\text{eff}}$. We have clarified this procedure in the supplementary information.

6) On pg 3 “state of the art PLD grown in-plane magnetized YIG//GGG at “such resonant” conditions. Can authors provide details (how thick, what freq)? If there is a reference, also cite that.

We have added a reference for YIG films grown using PLD with the same thickness that show a Gilbert damping parameter of $2.3 \cdot 10^{-4}$ (d’Allivy Kelly et al. APL 2013). This report is a self-citation.

7) Similarly, cite reference or give details about Pt/YIG//GGG

This point has also been addressed by d’Allivy Kelly et al. APL 2013. We can therefore guaranty that the experimental procedure is equivalent.

8) why was the SHE measured in an inplane film when the build up was all about PMA?

The FMR induced inverse spin Hall measurements is only sensitive to the in-plane component of the magnetization as the voltage is the result of a cross product:

$$\overrightarrow{V}_{\text{ISHE}} \propto \overrightarrow{J}_{\text{S}} \times \overrightarrow{M}$$

Therefore, we used a BiYIG film in-plane magnetized to perform the ISHE measurement and qualify the interface for spin transparency.

Figures:

- 1) Best to discuss “a-f” in order in the caption to step reader through the figure. “f” text labels are hard to read.

We have changed the labeling and modified the figure caption to make it more easy to read.

- 2) Add label “sGGG” to part “b”. Maybe show all values of x on GGG as “a” instead of repeating a sample. Why is Hsat so high? Seems larger than Hku would predict but it could be labeling of axis is just not clear. What is Hsat?

We thank the referee for its careful reading of the paper. This measurement was performed on a sample coming from a different growth batch than the one used for FMR. We have now corrected this discrepancy. We have redone the measurement on the same sample used in the FMR data.

- 3) Part “a” is not discussed in caption at all. Seems like it is backward, unless the film is down (green) and substrate up (blue)? Maybe a label in figure would help. “d” and “e” seem opposite from text and caption descriptions

We thank the referee for its careful reading of the paper. We have taking into account his suggestion and corrected our mislabeling.

Excellent paper that could use some work, but should definitely be published with the suggested updates.

We sincerely thank the referee for his help and his positive evaluation.

Second reply of referees: 22/05/2018

We are grateful for all three referees for their careful reading and their positive evaluations

Reviewer #1 (Remarks to the Author):

The authors reported the very low Gilbert damping coefficient of this PMA film, which is novel enough now to be considered publication. I can now recommend publication of this paper. I still suggest the authors to provide more structural information of the PMA origin, if possible in their final form of the submission.

We agree with the referee regarding the relevance of TEM measurement. We are still waiting to perform TEM experiments and hope to be able to report them as soon as possible.

Reviewer #2 (Remarks to the Author):

The manuscript has improved significantly and my concerns have been addressed.

I do recommend adding the following reference since it shows work on strain tuning PMA in the same BiYIG system that is being addressed here:

P. Sellappan et al “An integrated approach to doped thin films with strain tunable magnetic anisotropy: Powder synthesis, target preparation and pulsed laser deposition of Bi:YIG” Materials Research Letters, (2017) 5, 41-47. DOI: 10.1080/21663831.2016.1195779.

We thank the referee for his/her suggestion. We did know this paper but we chose not to cite it as most of the paper is focused on the target fabrication which is not our focus. Moreover the tensile strain in this paper is achieved due to the different thermal expansion coefficients between the substrate and the film and not due to an epitaxial strain induced elastic deformation of the unit cell (which is our case). We think that citing this article would be misleading on the origin of strain arising in our thin films.

Reviewer #3 (Remarks to the Author): This re-write is sufficient for publication. Accept as submitted.

We thank the referee for his/her positive evaluation.